# Energy Utilization of Torrefied Residue from Wine Production

**DOI:** 10.3390/ma14071610

**Published:** 2021-03-25

**Authors:** Barbora Tamelová, Jan Malaťák, Jan Velebil, Arkadiusz Gendek, Monika Aniszewska

**Affiliations:** 1Department of Technological Equipments of Buildings, Faculty of Engineering, Czech University of Life Sciences Prague, Kamycka 129, 165 21 Prague 6, Czech Republic; malatak@tf.czu.cz (J.M.); velebil@tf.czu.cz (J.V.); 2Department of Biosystems Engineering, Institute of Mechanical Engineering, Warsaw University of Life Sciences–SGGW, Nowoursynowska 164, 02-787 Warsaw, Poland; arkadiusz_gendek@sggw.edu.pl (A.G.); monika_aniszewska@sggw.edu.pl (M.A.)

**Keywords:** biochar, torrefaction, elemental analysis, energy properties, bioenergy

## Abstract

A significant amount of waste is generated in the food industry, which is both an environmental and an economic problem. The recycling of this waste has become an important area of research. The processing of grapes produces 20–30% of the waste in the form of grape pomace and stalks. This article assesses the fuel values of these materials before and after torrefaction. The input materials were grape pomace samples from the varieties Riesling (*Vitis vinifera* “Welschriesling”) and Cabernet Sauvignon (*Vitis vinifera* “Cabernet Sauvignon”) from the South Moravia region and stalks from the variety Welschriesling. The torrefaction process was performed using a LECO TGA 701 thermogravimetric analyzer under nitrogen atmosphere at set temperatures of 225 °C, 250 °C, and 275 °C. The residence time was 30 min. Elemental analysis, calorific value, and gross calorific value were determined for all samples. The analyses show a positive effect of torrefaction on fuel properties in the samples. Between temperatures 250 °C and 275 °C, the carbon content increased by 4.29 wt.%, and the calorific value increased with the increase in temperature reaching a value of 25.84 MJ·kg^−1^ at a peak temperature of 275 °C in the sample grape pomace from blue grapevine.

## 1. Introduction

In today’s world, owing to the issues of climate change and declining fossil-fuel supplies, a number of energy sector changes need to be addressed. For these reasons, renewable energy sources are coming to the forefront. The main benefit of renewable energy sources is the ability to reduce greenhouse gas emissions and environmental pollution.

One of the renewable energy sources is biomass. In recent years, waste biomass has become an alternative option for clean energy production. The use of biomass includes a wide range of potential thermochemical, physicochemical, and biochemical processes. The food industry generates a large amount of biodegradable waste, which is an environmental problem. Waste biomass processing by pyrolysis is considered a reliable method of producing high-quality renewable fuels [1]. Grapes are produced for direct consumption or for the production of natural wines. Furthermore, a wide range of products such as jams, jellies, vine juices, and raisins can be made from grapes. Oil can also be obtained from the seeds [2]. Vine growing is one of the most widespread activities in the world. World grape production is over 50 million tons per year, of which more than 20 million tons come from European producers [3]. Approximately 75% of grapes are processed to produce natural grape wine [2]. The European Union accounts for almost 70% of the world’s production of natural grape wine, with Italy, France, and Spain as the main European wine-producing countries [4]. Grape pomace is a solid organic waste from the wine industry. It is the residue from the pressing process. It contains seeds and husks, and, depending on the processing technology, it may also contain stems [5]; these residues represent a rich source of polysaccharides and phenolic compounds, both flavonoids and nonflavonoids [6,7]. Blue grape pomace has a large content of phenolic compounds such as anthocyanins, catechins, procyanidins, flavonol glycosides, phenolic acids, and stilbenes [8]. Pomace usually represents 20–30% of the weight of the unprocessed grapes [9]. Given this proportion, the wine industry produces millions of tons of residue, which represent an environmental and economic problem [10]. The amount of grape pomace produced will vary on the basis of the size of the winery and the methods used to make the wine. A small winery can process only a few tons of grapes, while large companies can process tens of thousands of tons [11]. Grape pomace is used abroad in various ways. In Italy, grappa is distilled [12,13]. The remaining grape pomace also serves as a good source of phytochemicals, including an array of phenolics, pigments, and antioxidants. These various ingredients are extracted using various techniques and the extracts have a number of applications [14,15]. Grape pomace can be used for composting [16] or as an animal feed [17]. Composting of grape pomace was studied by Paradelo et al. [16]. Potential difficulties for composting were identified, notably pH, which could inhibit the transition from mesophilic to thermophilic composting stages. During composting, most changes were observed to occur within the first 2 months. However, thermophilic conditions were not achieved, suggesting insufficient isolation of wastes during composting. Grape pomace also has significant potential as a bioenergy raw material (torrefaction, hydrothermal carbonization, combustion, gasification, pyrolysis). Pala et al. [6] investigated hydrothermal carbonization and torrefaction, Botelho et al. [18] also studied torrefied pomace, Encinar et al. [19] studied pyrolytic treatment, Lapuerta et al. [20] examined gasification, and Miranda et al. [21] evaluated the combustion of grape pomace. According to data from the Czech Statistical Office, a total of 90,000 tons of grapes were harvested in the Czech Republic in 2020, which represents a production of approximately 22,500 tons of grape pomace [22]. In the Czech Republic, the use of this raw material is very limited due to insufficient technology that would fully and effectively use the potential of grape pomace. A separate type of waste produced in winemaking is represented by the stalks. In some processing technologies, the stalks are separated from the grapes before processing. Stalks represent 3% to 6% of the total weight of the grapes [23]. The stalks have high contents of lignin, cellulose, and hemicelluloses, which all have relatively high carbon content [24]. Compared to other woody raw materials (wood chips, annual crops, etc.), the stems also contain higher amounts of condensed tannins [25]. Because of the large amount of stalks in the processing of grapes, the problem of their disposal must be optimized. This also presents an economic and environmental concern [26]. In practice, this results in their composting, incorporation into the soil, and other forms of biological treatment.

One way to utilize biomass from grape pomace and grape stalks and to reduce exhaust and carbon dioxide emissions is direct combustion [27,28], with fuel in the form of pellets or briquettes [29,30,31,32]. Another option is pretreatment with liquid hot water [33] or torrefaction.

Torrefaction as a thermal pretreatment technology is generally performed at temperatures varying from 200 °C to 300 °C under nitrogen atmosphere [34,35,36,37]. Nitrogen is the commonly used carrier gas to provide an inert atmosphere [38]. This pretreatment processing enhances the chemical and physical properties of raw materials [39]. Through the process, a uniform solid product known as torrefied biomass or biochar with low moisture content, high energy density, enhanced grindability, and hydrophobic properties is obtained [40,41,42,43,44]. The properties of biochar are influenced not only by the type of input biomass, but also by the conditions during the torrefaction process [45]. A review of past studies indicates that, when torrefaction is employed to pretreat biomass, some of the oxygen and hydrogen in raw biomass is consumed from the thermal degradation. Meanwhile, biomass experiences partial carbonization. The O:C and H:C ratios in torrefied biomass are reduced, thereby increasing its heating value [46,47]. The literature describes many studies related to the torrefaction of various materials, but there are very few publications describing the torrefaction of grape pomace and grape stalk.

The aim of this research was to assess the energy properties of grape pomace and stalks before and after torrefaction. Proximate, elemental analysis and calorific values were determined for these samples. Furthermore, stoichiometric parameters such as the theoretical amount of air for combustion or the amount of dry flue gas were determined for environmental impact. Therefore, the mass flow of fuel could be determined dependent on the desired heat output of the combustion apparatus.

## 2. Materials and Methods

### 2.1. Sample Preparation

Grape pomace was sampled at a winery near Prague as one whole batch immediately after pressing. The wine varieties were Riesling (*Vitis vinifera* “Welschriesling”) and Cabernet Sauvignon (*Vitis vinifera* “Cabernet Sauvignon”) from the South Moravia region. On the same date as the pomace, stalks were sampled for the Welschriesling variety. These waste samples were subjected to several treatments prior to a series of analyses. For each waste material, a sample was taken for determination of the original moisture. Next, the samples were air-dried without heating. Subsequently, the materials were milled to a size under 1 mm using a Retsch SM100 cutting mill (Retsch GmbH., Haan, Germany).

### 2.2. Preparation of Torrefied Samples

The LECO TGA 701 thermogravimetric analyzer (LECO Corporation, St. Joseph, MI, USA) was used for the preparation of torrefied samples. Materials were weighed in crucibles lined with aluminum foil and dried to constant weight. Then, an inert atmosphere using nitrogen was introduced before the torrefaction step. The constant nitrogen flow rate was 8.5 L·min^−1^. A total of three repetitions for each material were performed at each set temperature: 225 °C, 250 °C, and 275 °C for 30 min. During the torrefaction, weight loss was monitored as a function of time. The process was computer controlled, with the data recorded to the hard drive.

### 2.3. Sample Analysis

The determination of the elemental composition for carbon (C), hydrogen (H), nitrogen (N), and sulfur (S) was carried out in the LECO CHN628 + S analyzer (LECO Corporation, St. Joseph, MI, USA). The heat of combustion was determined in a LECO AC-600 isoperibolic calorimeter (LECO Corporation, St. Joseph, MI, USA). The samples were first pressed into pellets and then incinerated. The measurement was repeated at least three times to obtain reliable results. The net calorific value was determined through calculation from the results of elemental and proximate analysis of individual samples.

Furthermore, stoichiometric analyses were performed to calculate combustion characteristics. Results of stoichiometric calculations were converted to normal gas conditions (temperature T = 0 °C and pressure P = 101.325 kPa).

For each sample, the required mass flow of fuel to the combustion device was determined according to the required heat output of the combustion plant. The assumed thermal efficiency was 90% and the rated heat output was varied from 20 kW to 300 kW. The required mass flow of a fuel was calculated according to
(1)$${\dot{m}}_{pv}\;=\frac{Pk\times 100}{qn\times \eta \;}$$
where
${\dot{m}}_{pv}$ is the mass flow of fuel to the combustion chamber (kg·s^−1^), Pk is the boiler’s rated thermal output (W), qn is the fuel net calorific value (J·K^−1^), and η is the efficiency of the combustion device (%).

## 3. Results and Discussion

### 3.1. Elemental Analysis and Calorific Value

White grape pomace (GP-W), blue grape pomace (GP-B), and stalks (GS) were analyzed for moisture, ash, and elemental composition, as well as gross and net calorific values. Torrefaction was carried out at the temperatures 225 °C, 250 °C, and 275 °C for 30 min in all cases. Table 1 shows the analyses of each sample before and after torrefaction treatment. When comparing the moisture content of the original samples, the moisture content of the stalks was higher than that of the pomaces. Similar results were reported by González-Centeno et al. [48], where, for the 10 grape stalk samples examined, the moisture was on average 66.16 wt.%. The ash content in the dry GP-W marc sample was 6.13 wt.%, while, for GP-B, it was 5.00 wt.%. The ash content in the dry stalks was 8.23 wt.%. Botelho et al. [18] reported a dry basis ash value of 4.0 wt.%. When comparing with wood biomass, Prins et al. [49] determined dry ash content in beech wood at 1.2 wt.% and in larch at 0.1 wt.%. The ash content of the torrefied samples increased with increasing temperature. The highest ash content was reached at the temperature of 275 °C, which was 7.25 wt.% for pomaces and 13.60 wt.% for grape stalks.

The carbon content in the torrefied samples always increased, while the hydrogen and oxygen content decreased. This change occurred because volatile components containing higher proportions of H and O are released during torrefaction [50]. In the torrefied grape pomace, the carbon content increased by 3.25 wt.% at the temperature 225 °C compared to the original dry state. The largest increase in carbon content occurred in the GP-W sample between the torrefaction temperatures 250 °C and 275 °C, where the carbon content increased by 4.29 wt.%. The carbon content of GP-W at 275 °C reached 64.96 wt.%. The increase in carbon content of blue grape pomace at the temperature of 225 °C from the original state was lower. The carbon content increased only by 2.60 wt.%. With a further increase in temperature, the carbon content increased by about 4.00 wt.% with each 25 °C increase. In the sample GP-B at 275 °C, the carbon content reached 64.92 wt.%. Pala et al. [6] reported a carbon content in grape pomace of 55.92 wt.% after torrefaction at 225 °C with a residence time of 30 min. Chiou et al. [50] reported a carbon content of 51.44 wt.% after torrefaction at 230 °C for 40 min, going up to 56.43 wt.% after treatment at 260 °C for 40 min. The dry stalk samples had a lower carbon content (46.40 wt.%) compared to grape pomace samples. Deiana et al. [24] reported dry carbon content of 46.10 wt.%. The largest increase in carbon content (8.16 wt.%) for the grape stalk samples occurred between the original dry sample and the GS sample at 225 °C. As the process temperature increased, the carbon content increased to the final value of 63.45 wt.% in the GS sample at 275 °C. The hydrogen content of the monitored samples decreased slightly with torrefaction temperature. The sample of white grape pomace in the dry state contained 6.03 wt.% hydrogen. Similar contents in white grape pomace were measured by Burg et al. [51] at 5.82 wt.%. At higher torrefaction temperatures, hydrogen content decreased to 5.57 wt.% in the GP-W sample at 275 °C. The hydrogen contents in the blue grape pomace samples were similar. The hydrogen content in the blue grape marc sample was 6.13 wt.% in the dry original sample. Burg et al. [51] reported a hydrogen content in blue grape pomace of 5.96 wt.%. In the dry stalk samples, the hydrogen content was 5.47 wt.%. Again, it decreased at higher torrefaction temperatures to the lowest value of 4.32 wt.% in the GS sample at 275 °C.

The nitrogen content increased slightly during torrefaction. For white grape pomace samples, the nitrogen content was lower than that of blue grape pomace. The nitrogen content of the GP-W sample at 275 °C was 1.96 wt.% and that of the GP-B sample at 275 °C was 2.52 wt.%. The nitrogen content in the sample of stalks was 0.69 wt.% in the dry original sample. With increasing torrefaction temperature, the nitrogen content rose to 1.28 wt.% in the GS sample at 275 °C. Deiana et al. [24] reported a nitrogen content in dry grape stalks of 0.40 wt.%.

The oxygen content in all samples decreased during torrefaction. For the GP-W sample at 225 °C, the oxygen content decreased by 2.93 wt.% compared to the original dry sample. At higher temperatures, the oxygen content in white grape pomace decreased by almost 5.00 wt.%. The oxygen content in the GP-W sample at 275 °C was 20.22 wt.%, which is a relative decrease of 38.50% compared to the original dry sample. Chiou et al. [50] reported an oxygen content of 33.49 wt.% after torrefaction at 230 °C with a residence time of 40 min. At a torrefaction temperature of 260 °C and residence time of 40 min, the same author reported an oxygen content of 27.30 wt.%.

The trend in oxygen depletion in torrefied blue grape pomace was similar. Between the original dry sample and the GP-B sample at 225 °C, the oxygen loss was 3.05 wt.%. Furthermore, at higher temperatures, the oxygen content decreased by 4.45 wt.% between the GP-B samples at 225 °C and 250 °C, and by another 5.31 wt.% in the GP-B sample at 275 °C. At the highest temperature, the oxygen content in torrefied blue grape pomace, i.e., the GP-B sample at 275 °C, was reduced by 39.21 wt.% to a final oxygen content of 19.86 wt.%.

The dry stalk samples contained the most oxygen (39.18 wt.%). During torrefaction at 225 °C, the oxygen content decreased by 10.25 wt.%. The decrease in oxygen content was the most significant of all examined samples. The oxygen content further decreased with increasing temperature. The oxygen content of the GS sample at 275 °C was 17.32 wt.%. Overall, the oxygen content decreased by as much as 44.20 wt.% during torrefaction compared to the original sample.

The calorific values in all torrefied samples increased with increasing temperature. The net calorific value of samples of pomaces increased only around 1 MJ·kg^−1^ between the dry sample and torrefaction at 225 °C. Then, between the temperatures 225 °C and 250 °C, it increased by almost 2 MJ·kg^−1^ in both pomace samples. The increase in net calorific value between 250 °C and 275 °C was again almost 2 MJ·kg^−1^.

The net calorific value of the original dry stalks sample was 16.30 MJ·kg^−1^. During torrefaction at 225 °C this increased by 3.64 MJ·kg^−1^. After torrefaction at 250 °C, the net calorific value increased by 2.33 MJ·kg^−1^. At the highest torrefaction temperature, the net calorific value increased only by a further 1.71 MJ·kg^−1^. The net calorific value of torrefied stalks (GS) at 275 °C was 23.98 MJ·kg^−1^. Chen et al. [52] reported a net calorific value of 16.53 MJ·kg^−1^ for dry cotton stalk. They also reported a net calorific value of 18.85 MJ·kg^−1^ for torrefied cotton stalk at 250 °C with a residence time of 30 min.

### 3.2. Sample Weight Loss Depending on Torrefaction Temperature

Figure 1 shows the thermogravimetric graphs of mass loss and mass yields of individual samples over the course of torrefaction. The pomace samples from both white and blue grapes showed almost identical courses of weight loss over time. The graphs were slightly different from each other due to different elemental composition, especially in terms of ash content. Similar results were determined by Pala et al. [6], who used torrefaction temperatures of 250 °C and 300 °C. For the stalk samples, the mass yield at most temperatures was approximately 10% lower than that for pomace samples. Tamelová et al. [53] studied the weight loss of citrus peel during the torrefaction process, where the treatment temperatures were in the same range of 225–275 °C. The weight losses of citrus peel were approximately 10% higher compared to grape pomace samples. The higher weight loss of citrus peel can be explained by the higher ash content compared to grape pomace.

The results of the measurements were processed by regression statistical analysis to express the dependence of the mass yield on the processing time of individual samples of grape pomace and grape stalks.

The resulting regression statistical analysis of the mass recovery results is shown in Table 2. The resulting polynomial equations show analogous dependencies with a high value of reliability.

### 3.3. Stoichiometric Combustion

Table 3, Table 4 and Table 5 show the differences in stoichiometric combustion between the original pomace and stalk samples. These differences between the materials decreased with increasing torrefaction temperature, as was also observed in citrus wastes after this treatment [53]. In particular, the consumption of combustion air and the production of flue gas for the original samples showed significant differences of up to 20%. At the highest torrefaction temperature of 275 °C, the differences between processed samples were reduced to 6%. Due to the higher carbon concentration in the combustible matter, the pomace samples could reach higher concentrations of carbon dioxide in the flue gas than the stalks. These concentrations further increased after torrefaction treatment. Moreover, the nitrogen content in the samples increased with higher torrefaction temperature. This nitrogen can react with oxygen during combustion to form fuel-based nitrogen oxides in the flue gas and, thus, their emission concentrations would be higher [54,55,56]. The largest volume of nitrogen in the flue gas was determined for blue grape pomace after torrefaction at 275 °C at 10.91 m^3^·kg^−1^ N_2_ in dry flue gas. With increasing torrefaction temperature, the theoretical carbon dioxide concentration in the flue gas decreased, while the differences between pomaces and stalks are reduced. Overall, the stoichiometric analysis showed that torrefaction had a positive effect on the quality of the materials, as the stoichiometric combustion properties of the examined samples of pomaces and stalks were equalized with increasing treatment temperature.

Figure 2 shows the required mass flow of the materials to reach a nominal heat output of a combustion plant in the range of 20 to 300 kW. The results show large differences between the original pomace and stalk samples of up to 21%. This difference was reduced to only 6% after torrefaction at 275 °C. The required fuel mass flow decreased with increasing torrefaction temperature. This decrease stemmed mainly from the higher calorific value of the torrefied material.

As for the economic feasibility of torrefaction treatment, various studies found that torrefied biomass products might not be competitive against untreated biomass, as shown for wood pellets [39]. On the other hand, it was shown that torrefaction can theoretically be profitable with good process integration and under favorable market conditions [57,58].

Akbari et al. [59] assessed the economic feasibility of biochar production including biochar made from grape pomace by torrefaction at 250 °C for 30 min. Presuming the pomace would be a waste feedstock with zero market price, they calculated the cost of the resulting biochar at 2.29 USD/GJ.

## 4. Conclusions

Torrefaction treatment significantly affected the fuel properties of all investigated samples. The process was capable of producing better fuel compared to the original biomass, mainly by increasing the calorific value and reducing the oxygen content. With increasing process temperatures, the calorific value of all samples increased. The calorific value increased most in the sample of blue grape pomace, reaching a net calorific value of 25.84 MJ·kg^−1^ after torrefaction at 275 °C and a residence time of 30 min. The net calorific value of this sample would increase by about 2 MJ kg^−1^ with every increase in process temperature by 25 °C. The oxygen content of the blue grape pomace decreased by 39.21 wt.% after torrefaction compared to the original state. The weight loss during torrefaction showed a decreasing trend over time. Curves of weight loss of pomace from white and blue grapes showed almost the same trends. The highest weight loss was observed in a sample of grape stalks (almost 40 wt.%) after torrefaction at 275 °C. The energy utilization of the original samples from grape pomace and grape stalk without further treatment brings problems with regard to compliance with the combustion conditions and the greenhouse gas emission concentrations. Torrefaction stabilizes the material properties for fuel purposes and for storage. The concentration of carbon dioxide upon combustion changes significantly after torrefaction treatment. The maximum concentrations of carbon dioxide released into the atmosphere by combustion are then reduced, which contributes to reducing carbon emissions and the negative impact on the environment.

## Figures and Tables

**Figure 1 materials-14-01610-f001:**
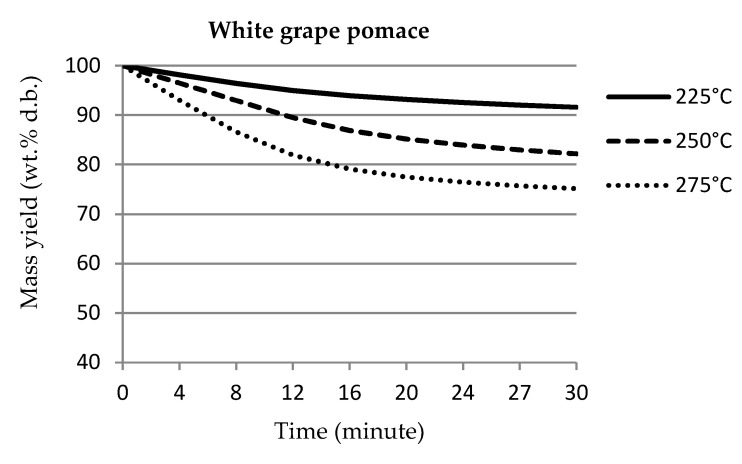
Mass loss curves of samples.

**Figure 2 materials-14-01610-f002:**
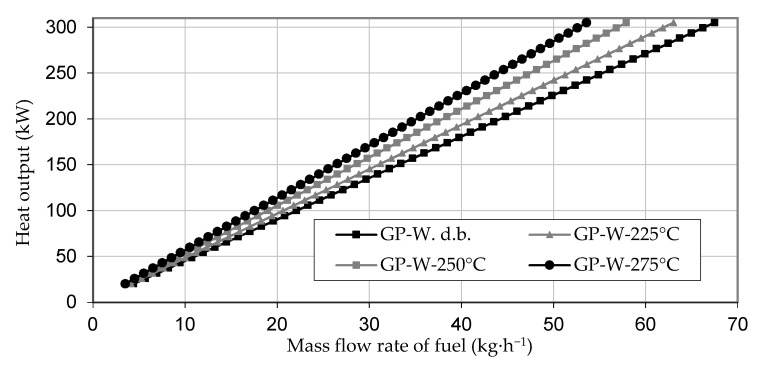
The mass flow rate of fuel to a combustion device for given heat output.

**Table 1 materials-14-01610-t001:** Composition of grape pomace (GP) and grape stalk (GS) before and after torrefaction treatment at varying temperatures and 30 min residence time.

Temp.	C	H	N	S	O	Ash	GCV	NCV
°C			%				MJ·kg^−1^	
GP-White								
Dry basis	53.29 ± 0.03	6.03 ± 0.02	1.64 ± 0.06	<0.02	32.88	6.13 ± 0.01	21.65 ± 0.03	20.33
225	56.54 ± 0.03	5.85 ± 0.01	1.75 ± 0.02	<0.02	29.95	5.88 ± 0.01	23.05 ± 0.06	21.78
250	60.67 ± 0.09	5.71 ± 0.02	1.86 ± 0.03	<0.02	25.11	6.62 ± 0.04	24.95 ± 0.04	23.71
275	64.96 ± 0.02	5.57 ± 0.03	1.96 ± 0.01	<0.02	20.22	7.25 ± 0.03	26.82 ± 0.02	25.61
GP-Blue								
Dry Basis	54.04 ± 0.09	6.13 ± 0.05	2.13 ± 0.06	<0.02	32.67	5.00 ± 0.03	22.17 ± 0.04	20.83
225	56.64 ± 0.08	6.03 ± 0.03	2.22 ± 0.03	<0.02	29.62	5.46 ± 0.05	23.44 ± 0.08	22.13
250	69.60 ± 0.10	5.93 ± 0.02	2.40 ± 0.03	<0.02	25.17	5.87 ± 0.04	25.24 ± 0.05	23.94
275	64.92 ± 0.02	5.84 ± 0.03	2.52 ± 0.06	<0.02	19.86	6.83 ± 0.03	27.12 ± 0.10	25.84
GS								
Dry Basis	46.40 ± 0.13	5.47 ± 0.01	0.69 ± 0.01	<0.02	39.18	8.23 ± 0.03	17.49 ± 0.03	16.30
225	54.56 ± 0.08	4.83 ± 0.03	1.03 ± 0.03	<0.02	28.93	10.61 ± 0.02	20.99 ± 0.08	19.94
250	59.92 ± 0.10	4.46 ± 0.01	1.25 ± 0.02	<0.02	21.96	12.38 ± 0.03	23.24 ± 0.09	22.27
275	63.45 ± 0.02	4.32 ± 0.03	1.28 ± 0.06	<0.02	17.32	13.60 ± 0.05	24.93 ± 0.04	23.98

GP-W, grape pomace from white grapevine; GP-B, grape pomace from blue grapevine; GS, grape stalk; GCV, gross calorific value; NCV, net calorific value.

**Table 2 materials-14-01610-t002:** Regression statistical analysis of mass recovery results.

Sample	Temperature	Polynomial Equation	Reliability Value
GP-W	225 °C	*y* = 0.1153*x*^2^ − 2.1784*x* + 102.01	*R*^2^ = 0.9988
	250 °C	*y* = 0.2546*x*^2^ − 4.781*x* + 104.74	*R*^2^ = 0.9987
	275 °C	*y* = 0.5295*x*^2^ − 8.2311*x* + 107.22	*R*^2^ = 0.9940
GP-B	225 °C	*y* = 0.0726*x*^2^ − 1.5408*x* + 101.54	*R*^2^ = 0.9993
	250 °C	*y* = 0.1427*x*^2^ − 3.4018*x* + 103.59	*R*^2^ = 0.9984
	275 °C	*y* = 0.4115*x*^2^ − 7.2972*x* + 107.47	*R*^2^ = 0.9974
GS	225 °C	*y* = 0.3314*x*^2^ − 5.4592*x* + 104.03	*R*^2^ = 0.9839
	250 °C	*y* = 0.6584*x*^2^ − 10.501*x* + 108.69	*R*^2^ = 0.9947
	275 °C	*y* = 0.9073*x*^2^ − 13.17*x* + 110.61	*R*^2^ = 0.9819

GP-W, grape pomace from white grapevine; GP-B, grape pomace from blue grapevine; GS, grape stalk.

**Table 3 materials-14-01610-t003:** Stoichiometric amount of air and specific productions of flue gas components from combustion grape pomace from white grapevine (GP-W); d.b., dry basis.

Parameter	Parameter	Unit	GP-W d.b.	GP-W 225 °C	GP-W 250 °C	GP-W 275 °C
L _min_	Stoichiometric Volume of Air for Complete Combustion	(m^3^·kg^−1^)	5.23	5.57	6.06	6.57
v_ssp min_	Stoichiometric Volume of Dry Flue Gas	(m^3^ kg^−1^)	5.08	5.41	5.87	6.35
v_CO2_	Stoichiometric Volume of CO_2_	(m^3^·kg^−1^)	0.99	1.05	1.13	1.21
v_H2O_	Stoichiometric Volume of H_2_O	(m^3^ kg^−1^)	1.11	1.12	1.14	1.17
v_N2_	Stoichiometric Volume of N_2_	(m^3^·kg^−1^)	8.59	9.14	9.95	10.78
CO_2_max	Concentration of Carbon Dioxide in Dry Flue Gas after Stoichiometric Combustion	(% vol.)	19.44	19.38	19.17	18.98

**Table 4 materials-14-01610-t004:** Stoichiometric amount of air and specific productions of flue gas components from combustion grape pomace from blue grapevine (GP-B); d.b., dry basis.

Parameter	Parameter	Unit	GP-B d.b.	GP-B 225 °C	GP-B 250 °C	GP-B 275 °C
L _min_	Stoichiometric Volume of Air for Complete Combustion	(m^3^·kg^−1^)	5.33	5.64	6.11	6.65
v_ssp min_	Stoichiometric Volume of Dry Flue Gas	(m^3^ kg^−1^)	5.18	5.47	5.91	6.41
v_CO2_	Stoichiometric Volume of CO_2_	(m^3^ kg^−1^)	1.01	1.05	1.13	1.21
v_H2O_	Stoichiometric Volume of H_2_O	(m^3^ kg^−1^)	1.13	1.14	1.17	1.21
v_N2_	Stoichiometric Volume of N_2_	(m^3^ kg^−1^)	8.75	9.26	10.03	10.91
CO_2_max	Concentration of Carbon Dioxide in Dry Flue Gas After Stoichiometric Combustion	(% vol.)	19.35	19.21	19.01	18.78

**Table 5 materials-14-01610-t005:** Stoichiometric amount of air and specific productions of flue gas components from combustion grape pomace from grape stalk (GS); d.b., dry basis.

Parameter	Parameter	Units	GS d.b.	GS 225 °C	GS 250 °C	GS 275 °C
L _min_	Stoichiometric Volume of Air for Complete Combustion	(m^3^ kg^−1^)	4.26	5.16	5.77	6.20
v_ssp min_	Stoichiometric Volume of Dry Flue Gas	(m^3^ kg^−1^)	4.19	5.04	5.62	6.02
v_CO2_	Stoichiometric Volume of CO_2_	(m^3^ kg^−1^)	0.86	1.01	1.11	1.18
v_H2O_	Stoichiometric Volume of H_2_O	(m^3^ kg^−1^)	0.97	0.97	0.98	1.00
v_N2_	Stoichiometric Volume of N_2_	(m^3^·kg^−1^)	6.99	8.46	9.46	10.17
CO_2_max	Concentration of Carbon Dioxide in Dry Flue Gas After Stoichiometric Combustion	(% vol.)	20.53	20.05	19.76	19.53

## Data Availability

Data sharing is not applicable to this article.

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
