# Peer review of "Energy Utilization of Torrefied Residue from Wine Production"

_materials, 2021, doi:10.3390/ma14071610_

Round 1
Reviewer 1 Report
Thank you for submitting this interesting work to materials. This is an interesting and timely study and I hope we can publish this soon. Below are a few comments that could further improve the quality of this work:
- Excellent study, I hope you did not just get wine redidues for your experiments
- do you also have total numbers on the grapte residue produced per year? so that we could upscale this
- what is the energy vaule of direct combustion - explain what ref 21-24 is about
- what is done at the moment with the waste in Europe, in Czche, your sutdy area? How much waste does an average wine producer accumulate per year?
- Maybe include a little figure with the material flows - grapes, wine, waste by mass to give a nice overview and a better idea of the problem
- table 2 is not required you can put this in the appendix or supplementary information if you really want to present it
- Table 6 is the most important one and you might want to consider presenting this in form of a figure to lighten the mood
Conclusions is also fine, this is a nice paper, what I miss is a little discussion chapter where you elaborate on the feasibility of actually using this process on an industrial scale.
Would it be feasible? economic? elaborate please.
Author Response
Thank you for your review. Below are the responses to your comments:
Point 1: What is the energy vaule of direct combustion - explain what ref 21-24 is about?
Response 1: Ref. 21-24: One way to utilize biomass from grape pomace and grape stalks and reduce exhaust and carbon dioxide emissions, is direct combustion [19, 20], with the fuel in the form of pellets or briquettes [21-24]. Ref. 21-24 was repaired in text. The torrefied material has a higher heating value than direct combustion.
Point 2: What is done at the moment with the waste in Europe, in Czche, your sutdy area? How much waste does an average wine producer accumulate per year?
Response 2: According to data from the Czech Statistical Office, a total of 90.000 tons of grapes were harvested in the Czech Republic in 2020, which represents a production of approximately 22.500 tons of grape pomace. Grape pomace is an effort to use abroad in various ways: In Italy, grappa is distilled, The remaining grape pomace also serves as a good source of phytochemicals, including an array of phenolics, pigments, and antioxidants. Grape pomace can be used as a animal feed, composting. But there was potential difficulties for composting were identified, notably pH, which could inhibit the transition from mesophilic to thermophilic composting stages. During composting, most changes were observed to occur within the first two months. However, thermophilic conditions were not achieved, suggesting insufficient isolation of wastes during composting. Grape pomace has also significant potential as a bioenergy raw material (torrefaction, hydrothermal carbonization, combustion, gasification, pyrolysis). More detailed text has been added to the introduction.
Point 3: Table 2 is not required you can put this in the appendix or supplementary information if you really want to present it
Response 3: We thoroughly thought about the suitability of table 2, we decided to keep it.
Point 4: Table 6 is the most important one and you might want to consider presenting this in form of a figure to lighten the mood.
Response 4: We changed the table 6 to figure.
Point 5: Conclusions is also fine, this is a nice paper, what I miss is a little discussion chapter where you elaborate on the feasibility of actually using this process on an industrial scale.
Would it be feasible? economic? elaborate please.
Response 5: A discussion of the economic feasibility was added at the end of the discussion section.
Reviewer 2 Report
The manuscript of Barbora Tamelová et al. deals with Energy Utilization of Torrefied Residue from Wine Production. The research is relevant to the Materials journal. However, some informations in the manuscript need could be improved, For this, the following MINOR REVISIONS are suggested.
Please specify the description of the torrefaction process.
What are the reasons for the differences in the amount of mass loss during the torrefaction process of different types of analyzed wastes?
Does the fragmentation of the grape pomace and stalks affect the material obtained after torrefaction?
Author Response
Thank you for your review. Below are the responses to your comments:
Point 1: Please specify the description of the torrefaction process.
Response 1: We added a description of the process to the methodology and introduction.
Point 2: What are the reasons for the differences in the amount of mass loss during the torrefaction process of different types of analyzed wastes?
Response 2: The curves are shifted due to different elemental composition, especially in the moisture of individual samples and ash content. We also compared it with other waste. This was added to text section 3.3.
Point 3: Does the fragmentation of the grape pomace and stalks affect the material obtained after torrefaction?
Response 3: Yes, the fragmentation of the grape pomace and stalk can affect the material obtained after torrefaction. In this study the materials were milled to size under 1 mm using a Retsch SM100 cutting mill.
Reviewer 3 Report
This paper is concerned with the fuel values through torrefaction treatment for grape waste. The topic of this study is interesting but there are several limitations:
- How is your research different from other similar studies? There are studies related to Torrefaction, but what is the difference compared to those studies?
- Are there any existing ways to deal with grape waste? It is worth mentioning that the existing method is a problem and it would be nice to show that this study can solve this problem.
- This research is said to help reduce carbon dioxide. What is the reduction criteria? Carbon dioxide is also likely to be generated in the process of Torrefaction. In addition, Landfill or other methods seem to be more helpful in reducing carbon dioxide.
Author Response
Thank you for your review. Below are the responses to your comments:
Point 1: How is your research different from other similar studies? There are studies related to Torrefaction, but what is the difference compared to those studies?
Response 1: Research differs by input materials and specified process temperatures of 225 °C, 250 °C and 270 °C.
Point 2: Are there any existing ways to deal with grape waste? It is worth mentioning that the existing method is a problem and it would be nice to show that this study can solve this problem?
Response 2: Grape pomace can be composting but there is potential difficulties for composting were identified, notably pH, which could inhibit the transition from mesophilic to thermophilic composting stages. During composting, most changes were observed to occur within the first two months. However, thermophilic conditions were not achieved, suggesting insufficient isolation of wastes during composting. Grape pomace has potential as a bioenergy raw materiál. Grape pomace has the potential as a bioenergy raw material, which could solve the problem of large amounts of waste. This was added to the introduction.
Point 3: This research is said to help reduce carbon dioxide. What is the reduction criteria?
Carbon dioxide is also likely to be generated in the process of Torrefaction. In addition,
Landfill or other methods seem to be more helpful in reducing carbon dioxide.
Response 3: The comment was meant in the sense that biomass derived fuels will generally have better carbon footprint compared to fossil fuels. However, this study was not aimed at evaluating the carbon footprint, so we decided to omit this mention. By landfilling the material, a significant amount of CO2 is also released, as well as methane in the landfill gas, which can be energetically utilized. However, landfilling biodegradable wastes is essentially banned in the European union, so we did not see that as a comparable option.
Carbon dioxide, as well as other gases and volatiles are released during torrefaction. In torrefaction, a share of carbon is lost from the sample in the form of gases and vapours. In our tests, this was around 9 % in the most severe reaction conditions for pomaces and as much as 14 % for the grape stalks. Our test apparatus did not enable us to find the composition of this fraction, neither the share of true gases, which would mainly contain CO2 and CO.
Round 2
Reviewer 1 Report
Thanks this is fine with me now
Reviewer 3 Report
The revised manuscript was greatly improved, and the questions were well answered and reflected in the manuscript. Therefore I would like to recommend accept.